# Influence of femoral anteversion angle and neck-shaft angle on muscle forces and joint loading during walking

Hans Kainz[1]*, Gabriel T. Mindler[2,3], Andreas Kranzl[3,4]

**1** Centre for Sport Science and University Sports, Department of Biomechanics, Kinesiology and Computer Science in Sport, Neuromechanics Research Group, University of Vienna, Vienna, Austria, **2** Department of Pediatric Orthopaedics, Orthopaedic Hospital Speising, Vienna, Austria, **3** Vienna Bone and Growth Center, Vienna, Austria, **4** Laboratory for Gait and Movement Analysis, Orthopaedic Hospital Speising, Vienna, Austria

\* hans.kainz@univie.ac.at

**Data Availability Statement:** Data related to this study is freely available on https://simtk.org/projects/bone_gait_load.

**Funding:** Open access funding provided by University of Vienna. The funders had no role in

## Abstract

Femoral deformities, e.g. increased or decreased femoral anteversion (AVA) and neck-shaft angle (NSA), can lead to pathological gait patterns, altered joint loads, and degenerative joint diseases. The mechanism how femoral geometry influences muscle forces and joint load during walking is still not fully understood. The objective of our study was to investigate the influence of femoral AVA and NSA on muscle forces and joint loads during walking. We conducted a comprehensive musculoskeletal modelling study based on three-dimensional motion capture data of a healthy person with a typical gait pattern. We created 25 musculoskeletal models with a variety of NSA (93˚-153˚) and AVA (-12˚-48˚). For each model we calculated moment arms, muscle forces, muscle moments, co-contraction indices and joint loads using OpenSim. Multiple regression analyses were used to predict muscle activations, muscle moments, co-contraction indices, and joint contact forces based on the femoral geometry. We found a significant increase in co-contraction of hip and knee joint spanning muscles in models with increasing AVA and NSA, which led to a substantial increase in hip and knee joint contact forces. Decreased AVA and NSA had a minor impact on muscle and joint contact forces. Large AVA lead to increases in both knee and hip contact forces. Large NSA (153˚) combined with large AVA (48˚) led to increases in hip joint contact forces by five times body weight. Low NSA (108˚ and 93˚) combined with large AVA (48˚) led to two-fold increases in the second peak of the knee contact forces. Increased joint contact forces in models with increased AVA and NSA were linked to changes in hip muscle moment arms and compensatory increases in hip and knee muscle forces. Knowing the influence of femoral geometry on muscle forces and joint loads can help clinicians to improve treatment strategies in patients with femoral deformities.

study design, data collection and analysis, decision to publish, or preparation of the manuscript.

**Competing interests:** The authors have declared that no competing interests exist.

## Introduction

Torsional deformities of lower limbs are common in patients with and without neurological disorders [1, 2], and are a frequent reason for consultation in pediatric orthopedics [3]. Many idiopathic torsional deformities seen in daily practice are minor and have little clinical significance. However, excessive malalignment can lead to an altered gait pattern, increased risk of falls, functional limitations, overuse injuries, joint pain and increased risk for clinical problems such as osteoarthritis [4–8].

The femoral anteversion and neck-shaft angle are two important anatomical features of the femur [9]. Anteversion angle (AVA) is the angle in the transverse plane by which the neck of the femur deviates forwards from the knee axis of the femoral condyles. The neck-shaft angle (NSA) is the angle between the neck and the shaft of the femur. In typically developing children, femoral AVA decreases from approximately 40˚ at birth to 15˚ at skeletal maturity, whereas the NSA decreases from 140˚ to 125˚. Torsional femoral deformities are defined as an increased or decreased AVA of a patient compared to age-matched typically developing children and occur in children with and without neurological disorders. In some individuals the AVA and NSA barely change during childhood and therefore with time develop large differences to the values of typically developing children [1, 10, 11]. In healthy adults, typical AVA and NSA are independent of each other with average values between 10˚ and 20˚ and between 124˚ and 136˚, respectively [12, 13].

Musculoskeletal modelling has been used to increase our insights in torsional deformities for more than two decades. Early studies showed that increasing the AVA decreased the abduction moment arm of the gluteus medius muscles but internal rotation of the hip restores the moment arm [14]. Furthermore, it has been shown that de-rotation osteotomies barely change the lengths of the hamstrings, gracilis and adductor muscles [15]. In children with cerebral palsy and internally-rotated gait the medial hamstrings, adductor brevis and gracilis muscles have a negligible internal rotation moment arm and therefore are unlikely to contribute to the internal rotated gait [16]. Increasing AVA shifted the moment arms from the hamstrings and adductors towards external rotation and therefore these muscles are unlikely to cause the internal rotated gait in children with cerebral palsy [17]. Further studies showed that a patient-specific gait pattern in children with cerebral palsy and increased AVA beneficially increased the ability of the gluteus medius and maximus to extend the hip and knee, whereas the potential of the hamstrings to extend the hip was decreased [18]. Furthermore, the patient-specific gait pattern in children with increased AVA reduces hip loading [19]. Heller et al. [20] showed that increasing the AVA increases hip joint contact forces when tracking subject-specific motion capture data of four patients with a total hip arthroplasty, which was confirmed by recent simulation studies [21–24]. Increased hip and patellofemoral loading was found in children with increased AVA and normal foot progression angle [25], whereas decreased hip and knee loads were observed in children with increased AVA and internal foot progression angle compared to healthy control participants [26]. In summary, previous studies showed that increased AVA alters the moment arms of certain muscles and leads to increased hip and knee joint loads, which can be compensated with an altered gait pattern. However, it remains unclear how the altered moment arms influence muscle forces and the generated moments by each muscle and therefore alter joint loads. Furthermore, it is unknown how the NSA influences and contributes to the altered muscle forces and joint loading.

The aim of our study was to comprehensively investigate the influence of femoral AVA and NSA on muscle forces and joint loads during walking. Investigating the influence of femoral geometry on joint loads is challenging with conventional approaches because each person has a subject-specific femoral AVA and NSA, and walks with an individual walking pattern, i.e.

joint angles and walking velocity. Hence, numerous variables would affect the estimated joint loads. To overcome this limitation, we decided to conduct a series of 'what-if' simulations, which enabled us to keep all variables constant and isolate the impact of altered femoral geometry on muscle forces and joint loads. We created 25 musculoskeletal models with a variety of NSA and AVA. For each model we calculated moment arms, muscle forces and joint loads based on the motion capture data of a person with a typical gait pattern. We hypothesized that the increased joint loads in models with increasing AVA are caused by increased co-contraction of hip and knee spanning muscles. Based on previous research [21], we furthermore hypothesized that models with increasing NSA will increase co-contraction and joint loads. Knowing the influence of femoral geometry on muscle forces and joint loads during walking could help clinicians to improve treatment strategies in patients with femoral deformities.

## Methods

### Musculoskeletal models

The gait2392 musculoskeletal OpenSim model [27, 28] was used as the reference model for our simulations. The model included three degrees of freedom at each hip joint, one degree of freedom at each knee and ankle joint and three degrees of freedom between the pelvis and torso segment. Furthermore, the model included 92 muscle-tendon units, representing the muscles in the lower extremities and torso. The femur in the reference model had an NSA of 123˚ and an AVA of 18˚. To investigate the impact of proximal femoral geometry on muscle forces and joint loads in a systematic way we created 24 additional models with varying NSA (±30˚ in 15˚ steps, NSA of 93˚, 108˚, 123˚, 138˚ and 153˚) and AVA (±30˚ in 15˚ steps, AVA of -12˚, 3˚, 18˚, 33˚ and 48˚). Both variables (AVA and NSA) were varied separately and together compared to the values of the reference model. The recently developed and published Torsion Tool [29] to personalize bony geometries in OpenSim models was used to generate the additional models. Briefly described, the tool changes the vertices of the femur based on pre-defined boxes to match the chosen NSA and AVA. This procedure alters all the muscle origin and insertion points within the boxes. The tool altered the proximal femoral geometry and therefore did not influence the anatomical coordinate system of the femur. Furthermore, all models were scaled in the same way. Hence, all our models had the same anatomical reference systems and segment dimensions, and therefore led to identical joint angles and moments [30, 31].

### Motion capture data

All simulations were based on three-dimensional gait analysis data, i.e. marker trajectories and ground reaction forces, obtained during barefoot walking on an instrumented walkway from a healthy person (mass: 73.1 kg, height: 1.71 m, walking velocity: 1.41 m/s) without any known abnormalities that could have altered the participant's gait pattern. Data was collected with a modified Cleveland marker set for the lower extremities and a Plug-in Gait marker set for the upper extremities [32]. All methods were carried out in accordance with the relevant guidelines and regulations. The used dataset for our simulations was part of a bigger project approved by our local ethics committee (Number: EK20/2022, Ethikkommission der Wiener Häuser der Vinzenz Gruppe, Vienna, Austria) for capturing reference data for the gait laboratory database of the Orthopedic Hospital Speising (Vienna, Austria). Signed written informed consent to use the collected data for the laboratory reference database and scientific studies was obtained from the participant prior to the measurements. AK selected a random data set for this study. He was the only person who had access to information that could identify the participant. All further processing and musculoskeletal simulations were performed based on the anonymized data set.

## Musculoskeletal simulations

Each musculoskeletal model was scaled to the anthropometry of our participant based on the location of surface markers and estimated joint centres [33]. Optimal fiber lengths and tendon slack lengths of each muscle were optimized to fit to each scaled model using the Matlab tool developed by Modenese et al. [34]. Muscle activations of the reference model (NSA of 123˚ and AVA of 18˚) led to unrealistic high values, i.e. 100% activation for several muscles. Hence, the maximum isometric force of each muscle was multiplied by two to allow the generation of realistic muscle activation waveforms with the reference model, i.e. avoid plateaus of 100% muscle activation. Increasing maximum isometric muscle forces of the gait2392 model is a common practice when analyzing movements of healthy, young adults [35, 36] because the original force values are based on data from elderly specimens and therefore are not representative for our participant. Inverse kinematics and inverse dynamics were used to calculate joint angles and moments, respectively. Muscle forces were estimated using static optimization while minimizing the sum of squared muscle activations and accounting for the muscle force-length-velocity relationship [37]. Afterwards, joint reaction load analysis [38] was performed to calculate hip, knee, and ankle joint contact forces. OpenSim's analyze tool was used to obtain the muscles' moment arms and muscle-tendon lengths for each model. All simulations were performed in OpenSim 4.2.

## Data analyses

All simulation results were normalized to 100% of the gait cycle. Additionally, joint moments were normalized to body mass, and muscle forces and joint contact forces were normalized to body weight. The moment each muscle generates was calculated by multiplying the muscle forces with the moment arm in each anatomical plane for each time frame during the gait cycle. The functional role of each muscle was defined by the muscle's moment arm during each frame of the gait cycle [39]. To quantify agonist ($M_{agonist}$) and antagonist muscle moments ($M_{antagonist}$) the sum of muscle moments for each anatomical plane (e.g. hip flexion/extension) and direction (e.g. sum of all positive values for hip flexion moments; sum of all negative values for hip extension moments) was calculated [40]. These muscle moments are referred to as hip flexors and extensors, hip abductors and adductors, hip internal and external rotators, knee flexors and extensors muscle moments in our study. Similar to previous studies [40], we calculated the co-contraction index (CCI) based on the muscle moments to quantify the amount of co-contraction (Eq 1). CCI values of 0, 1 and -1 indicate full co-contraction, only antagonist activation, and only agonist activation, respectively. Considering that the CCI does not give us a value for the magnitude of co-contraction, we additionally compared the muscle moments between the corresponding agonist and antagonist muscle groups. To address our hypotheses, i.e. increased NSA and AVA increase co-contraction and joint contact forces, we used a multiple regression analysis to predict the mean muscle activations, muscle moments, CCI, and joint contact forces during the stance phase of gait based on the femoral geometry, i.e. NSA and AVA. Statistical significance was set at alpha = 0.05. Additionally, we used descriptive statistics to compare joint angles, joint moments, muscles' moment arms and muscles' lengths between the different models.

$$CCI = \begin{cases} 1 - \frac{M_{agonist}}{M_{antagonist}}, \; if \; M_{antagonist} > M_{agonist} \\ \frac{M_{antagonist}}{M_{agonist}} - 1, \; otherwise \end{cases} \tag{1}$$

## Results

### Joint angles and moments

Joint angles and moments of our participant were identical between the different models and comparable to previous studies based on healthy individuals [41] (Fig 1).

### Joint contact forces

Hip and knee joint contact forces increased with increasing NSA and AVA (Figs 2 and 3). The femoral geometry significantly predicted hip ($R^2 = 0.86$, $p<0.001$) and knee ($R^2 = 0.60$, $p<0.001$) JCF. Both variables, i.e. AVA and NSA, added statistically significantly ($p<0.001$) to the prediction.

### Co-contraction and muscle moments

Increased AVA increased hip flexor/extensor co-contraction and increased hip flexor, hip extensor, knee flexor and knee extensor muscle moments (Figs 4 and 5). Increased NSA increased knee flexor and knee extensor muscle moments and decreased hip internal and external muscle moments.

The femoral geometry significantly predicted hip flexor/extensor CCI ($R^2 = 0.92$, $p<0.001$), hip ab-/adductor CCI ($R^2 = 0.47$, $p<0.001$), hip internal/external rotator CCI ($R^2 = 0.84$, $p<0.001$) and knee flexor/extensor CCI ($R^2 = 0.27$, $p = 0.033$). Both variables, i.e. AVA and NSA, added statistically significantly ($p<0.05$) to the prediction of hip ab-/adductor CCI and hip internal/external rotator CCI, whereas only AVA was a significant predictor ($p<0.001$) for hip flexor/extensor CCI and only NSA was a significant predictor ($p<0.01$) for knee flexor/extensor CCI.

The femoral geometry significantly predicted all analyzed muscle moments ($R^2$ between 0.57 and 0.84, $p<0.001$ for all moments). Both variables, i.e. AVA and NSA, added statistically significantly ($p<0.05$) to the prediction of muscle moments except for hip internal and external muscle moments, where only the NSA was a significant predictor.

### Muscle activations

Increasing AVA increased muscle activation of all hip and knee spanning muscle groups (Fig 6). Increasing NSA altered muscle activations of all hip and knee spanning muscle groups except for hip flexor, hip external rotators and knee extensor muscles.

The femoral geometry significantly predicted muscle activations of all analyzed muscle groups ($R^2$ between 0.54 and 0.82, $p<0.001$ for all muscle groups). Both variables, i.e. AVA and NSA, added statistically significantly ($p<0.05$) to the prediction of muscle activations except for hip flexor, hip external rotators and knee extensor muscles, where only the AVA was a significant predictor.

### Moment arms and muscle forces

Average moment arms of hip extensor muscles increased with increasing AVA, whereas moment arms of hip flexor muscles barely changed (Fig 7). Both hip flexor and extensor muscle forces increased with increasing AVA but the increase in muscle forces was higher for hip flexor compared to hip extensor muscles (Fig 8).

Increased AVA decreased the hip abduction moment arms of the majority of hip abductor muscles (Fig 7), i.e. gluteus minimus, medius, and maximus, tensor fasciae latae, and piriformis. None of the hip abductor moment arms increased with increasing AVA. Increasing NSA led to an additional decrease in hip abduction moment arms. Both hip abductor and adductor

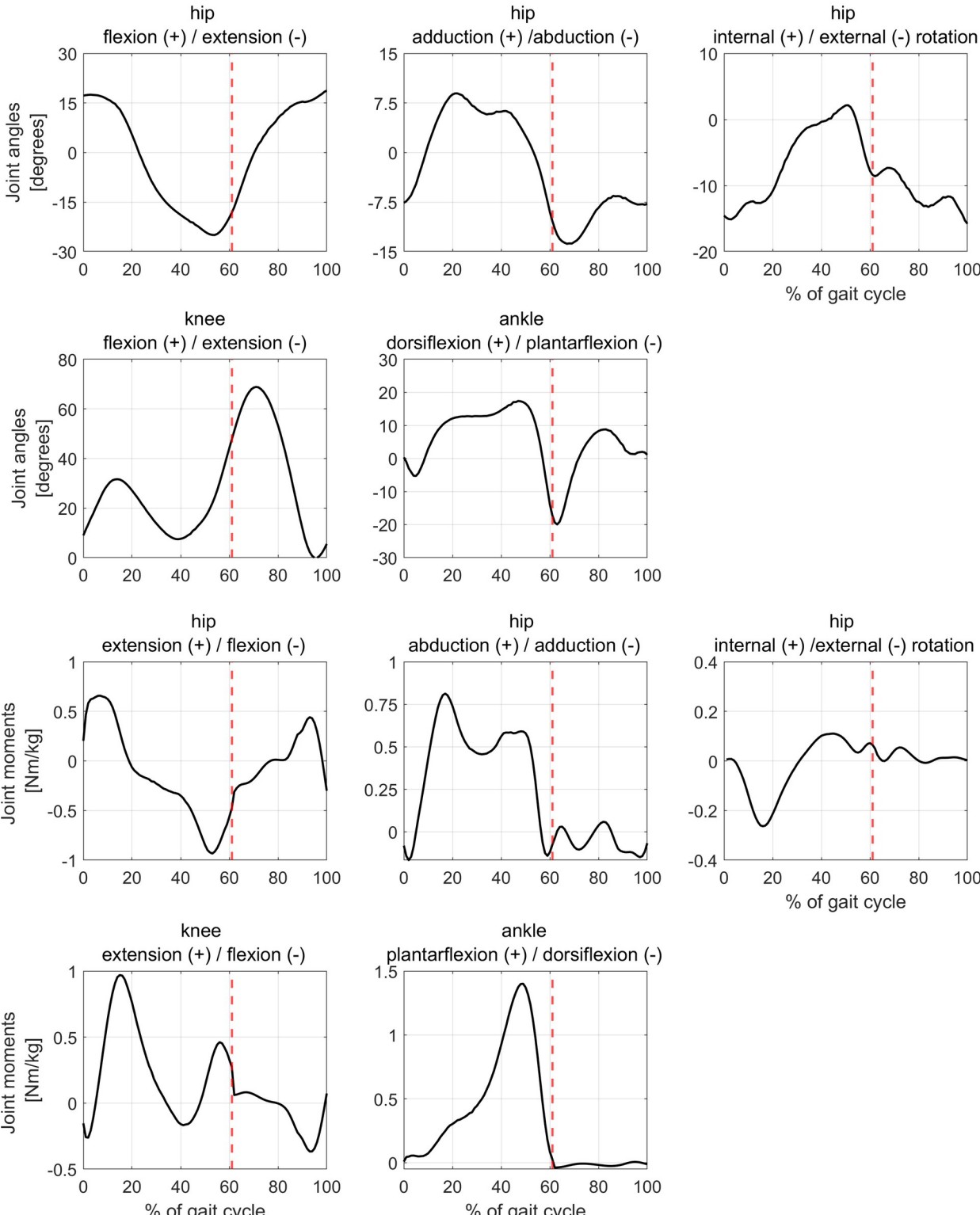

**Fig 1.** Joint kinematics (top two rows) and kinetics (bottom two rows). The obtained waveforms from all our models were identical. Red, dashed vertical lines indicate the end of the stance phase.

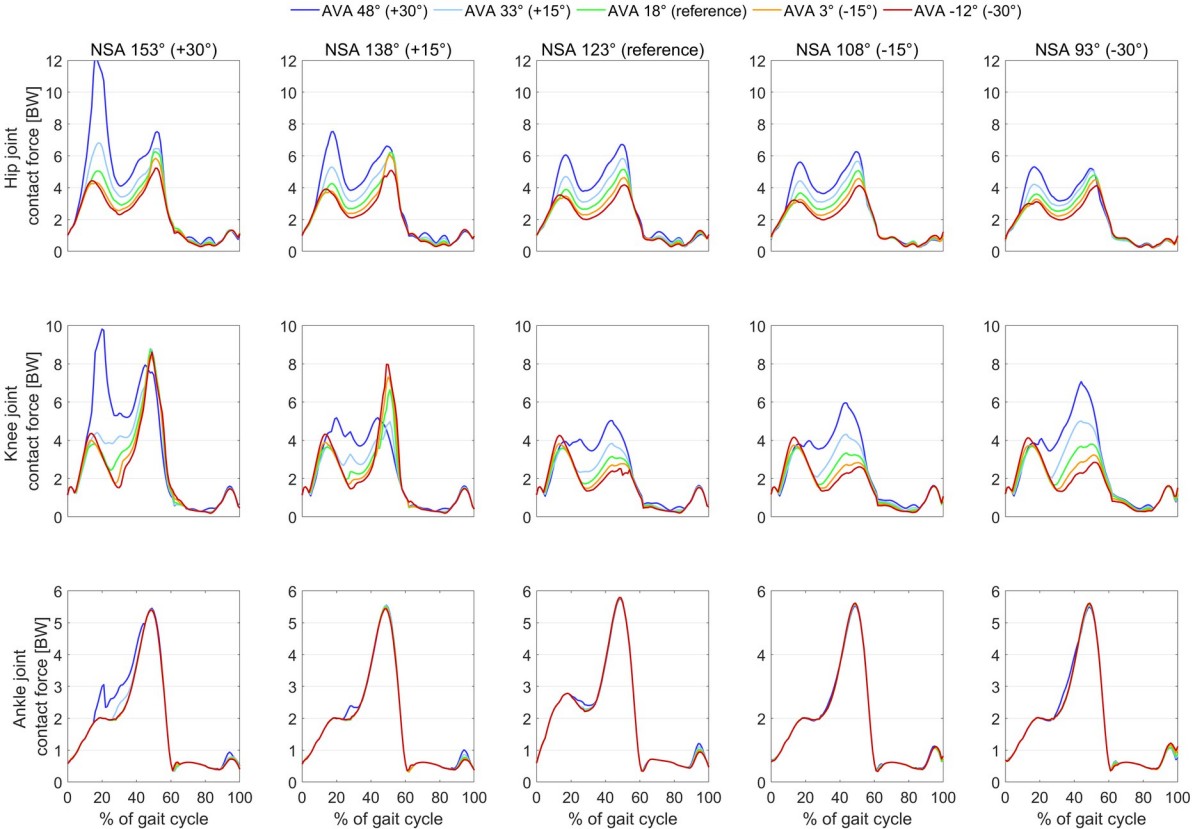

**Fig 2. Resultant hip, knee and ankle joint contact forces obtained with the different models.** The geometry (NSA and AVA) had a big impact on hip and knee joint contact forces. Large AVA combined with large NSA led to joint contact forces up to 12 times body weight. It is unlikely that a person would walk with such high joint contact forces. People with torsional deformities are able to decrease joint contact forces with a pathological gait pattern, e.g. in-toeing gait, which should be kept in mind when interpreting our findings.

muscle forces increased with increasing AVA but the increase in muscle forces was higher for hip abductor compared to hip adductor muscles (Fig 8).

Average moment arms of hip internal and external rotator muscles slightly decreased with increasing AVA, whereas muscle forces increased with increasing AVA. Increasing NSA decreased hip internal rotator moment arms (Figs 7 and 8). No change in knee flexion/extension moment arms were observed between models but increasing the AVA led to increased knee flexor and extensor muscle forces.

## Muscle-tendon length

For the majority of muscles, the mean muscle-tendon lengths during the stance phase of the gait cycle did not change with the altered femoral geometry (S1 Fig in S1 File). A small decrease in muscle-tendon length with increasing AVA was found for the gluteus maximus, quadratus femoris, gemellus and piriformis muscle.

## Verification of simulation results

The simulation from our reference model, i.e. AVA of 18˚ and NSA of 123˚, led to maximum hip, knee and ankle joint contact forces of 5.1 times body weight (BW), 3.6 BW, and 5.7 BW,

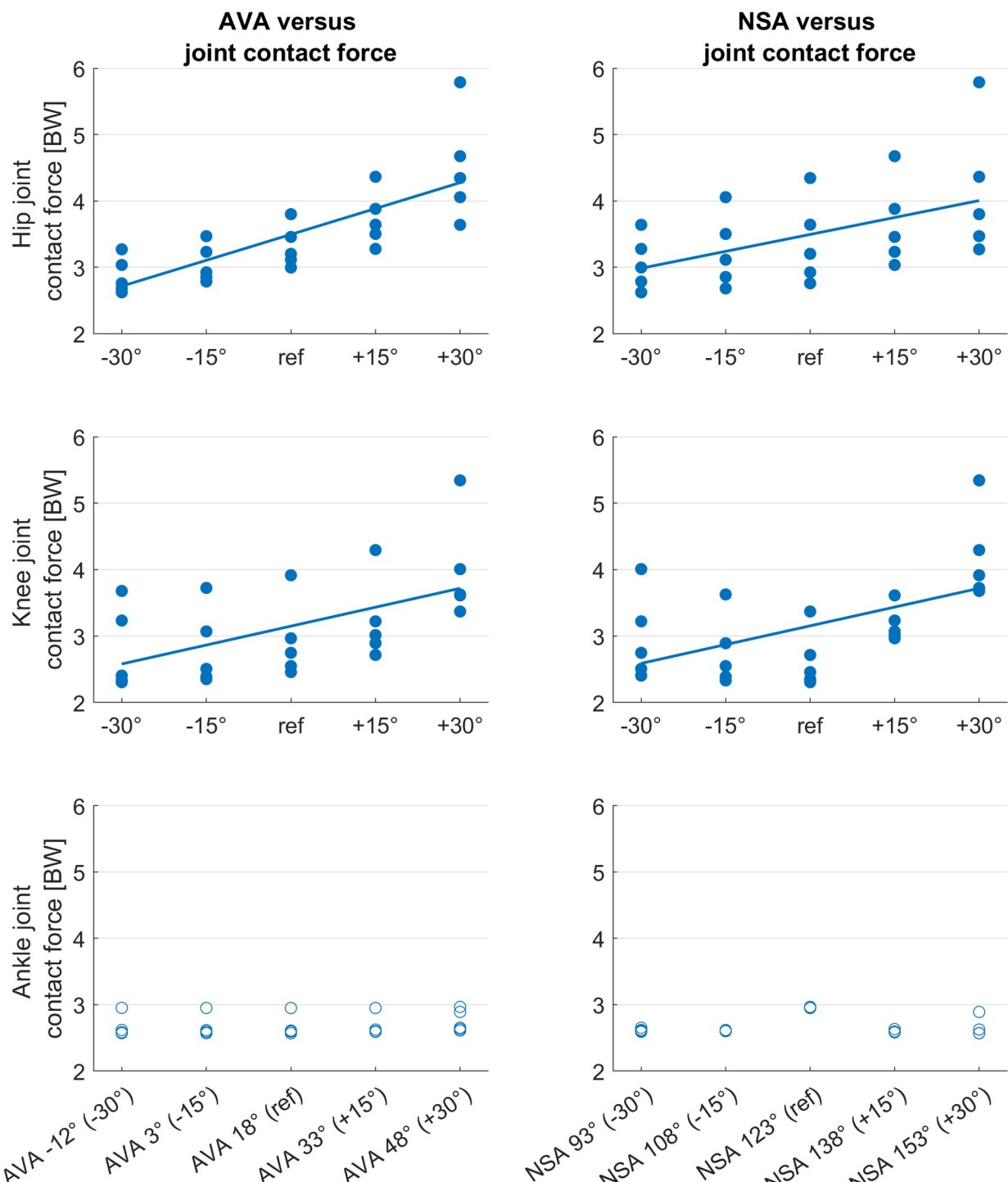

**Fig 3. Scatterplots showing the relationship between anteversion angle (AVA) and mean joint contact forces during the stance phase, and neck-shaft angle (NSA) and mean joint contact forces during the stance phase.** Filled circles indicate significant predictors (p<0.001) from the multiple regression analysis. BW = body weight. Straight lines are the regression lines obtained from the multiple regression analysis.

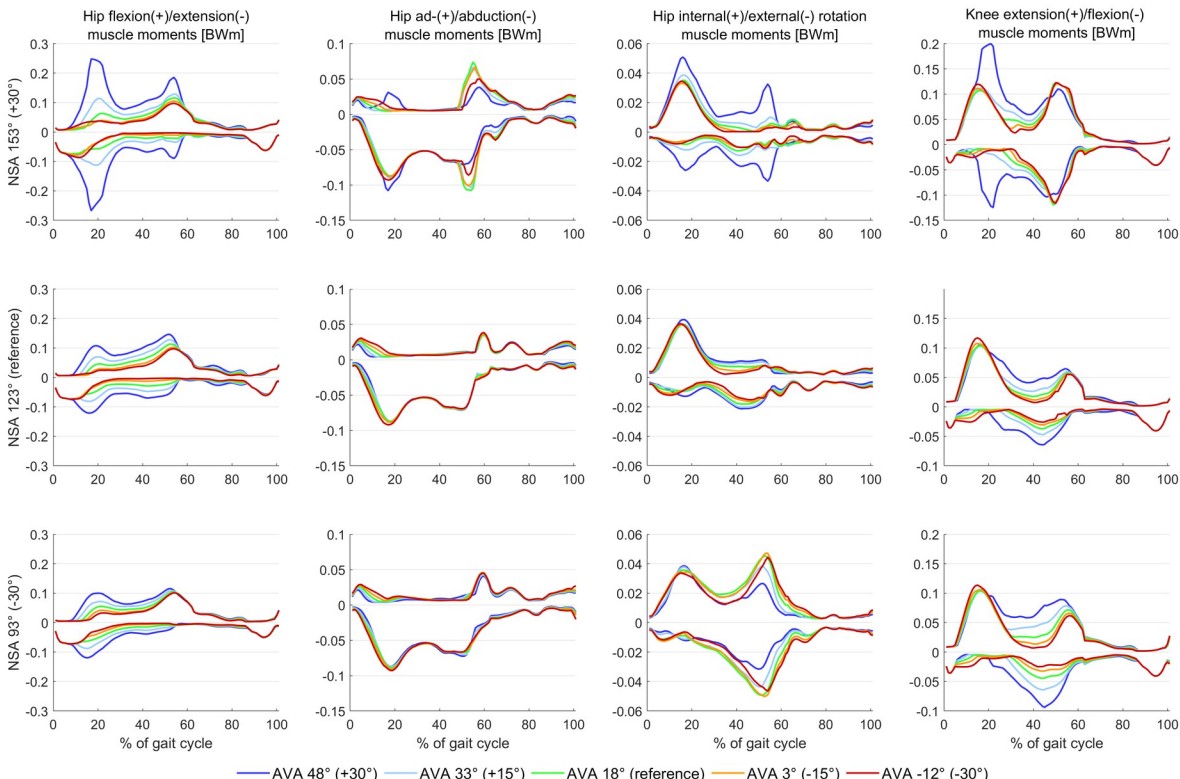

**Fig 4. Agonist and antagonist muscle moments obtained from models with different femoral geometry.** Both altered AVA and NSA had an influence on the obtained muscle moments. In the NSA 153˚ AVA 48˚ model, the increased first peak of the knee flexion moment was mainly caused by an increased muscle force and moment of the rectus femoris muscle.

respectively. The magnitude and shape of the contact force waveforms were in agreement with previous simulation studies [25, 42–44]. In-vivo measurements of joint loads based on instrumented implants showed lower hip and knee joint contact forces (maximum values of 3.2 BW) [45] compared to our simulations, potentially due to the different walking velocity [46, 47] between the elderly patients with a joint replacement (3–4 km/h) and our young, healthy participant (5.1 km/h). Muscle activations and forces from our simulations showed a reasonable agreement with experimentally measured electromyography signals [48, 49] and previously estimated muscle forces [39, 50] (supplementary material).

We plotted the moment arms and muscle length from all models, i.e. reference and deformed models, to verify that muscle-tendon kinematics is reasonable, i.e. does not lead to discontinuities during the walking pattern of our participant. For all models, dynamic muscle moment arms and muscle-tendon lengths showed smooth waveforms throughout the whole gait cycle of our participant.

## Discussion

We comprehensively investigated the influence of femoral AVA and NSA on muscle forces and joint loads during walking. In agreement with our hypotheses, we showed that the AVA and NSA are significant ($p<0.05$) predictors for co-contraction and muscle moments in hip and knee joint spanning muscles, and hip and knee joint contact forces. Increased iliacus, psoas, gluteus, tensor fasciae latae, and rectus femoris forces caused increased hip flexor and

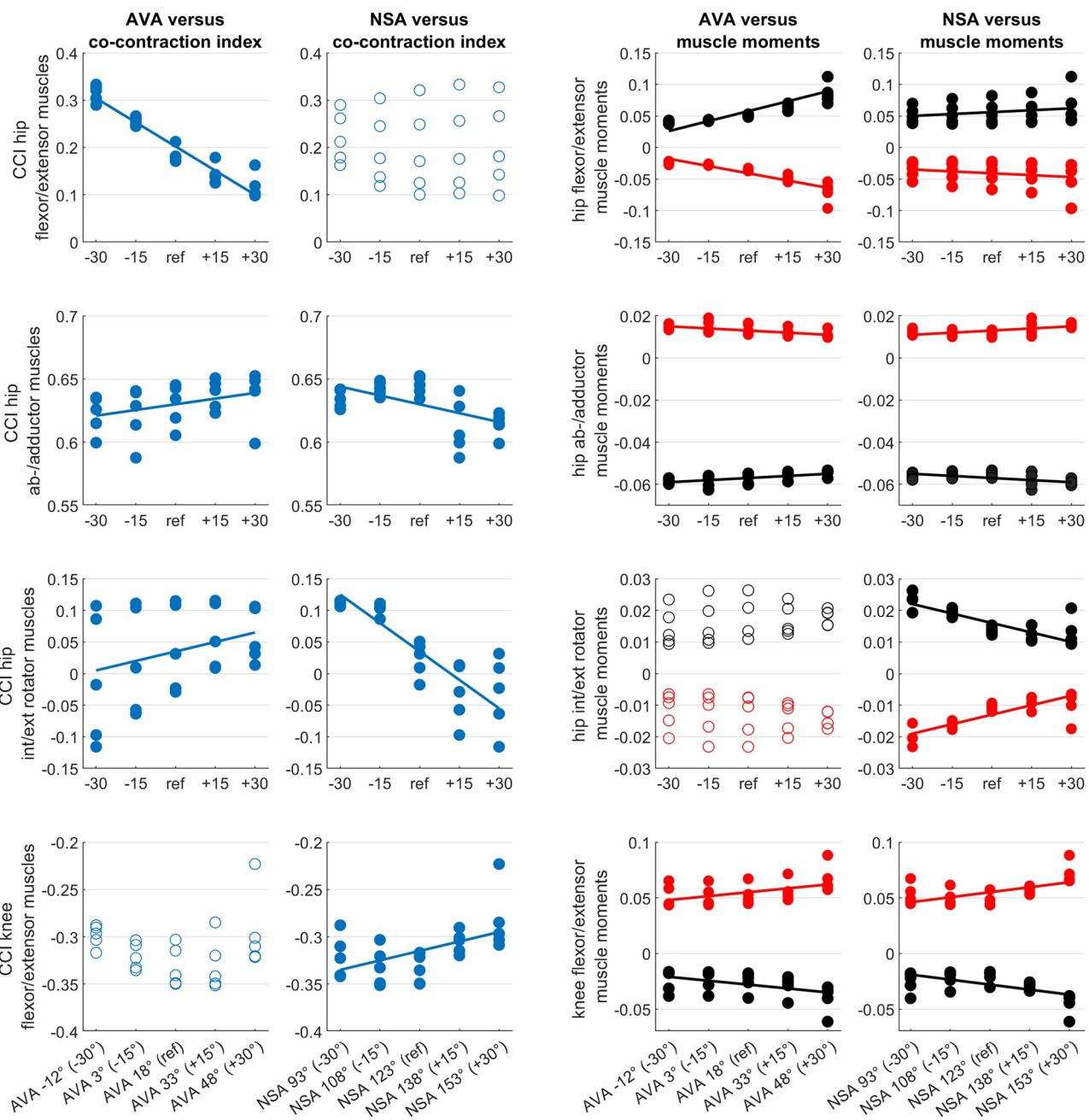

**Fig 5. Scatterplots showing the relationship between the femoral geometry (AVA and NSA) and mean co-contraction indices (CCI) and mean muscle moments during the stance phase of the gait cycle.** Filled circles indicate significant predictors (p<0.05) from the multiple regression analysis. Straight lines are the regression lines obtained from the multiple regression analysis. CCI values of 0 indicate full co-contraction. CCI values of 1 and -1 indicate only antagonist activation and only agonist activation, respectively.

extensor muscle moments and explain the increase in hip joint contact forces in models with increasing AVA. In models with increasing NSA, the majority of hip muscle moment arms decreased and muscle forces increased, which explains the increased hip joint contact forces despite similar or even decreasing agonist and antagonist muscle moments. Increased rectus femoris and gastrocnemius muscle forces were mainly responsible for the increased knee

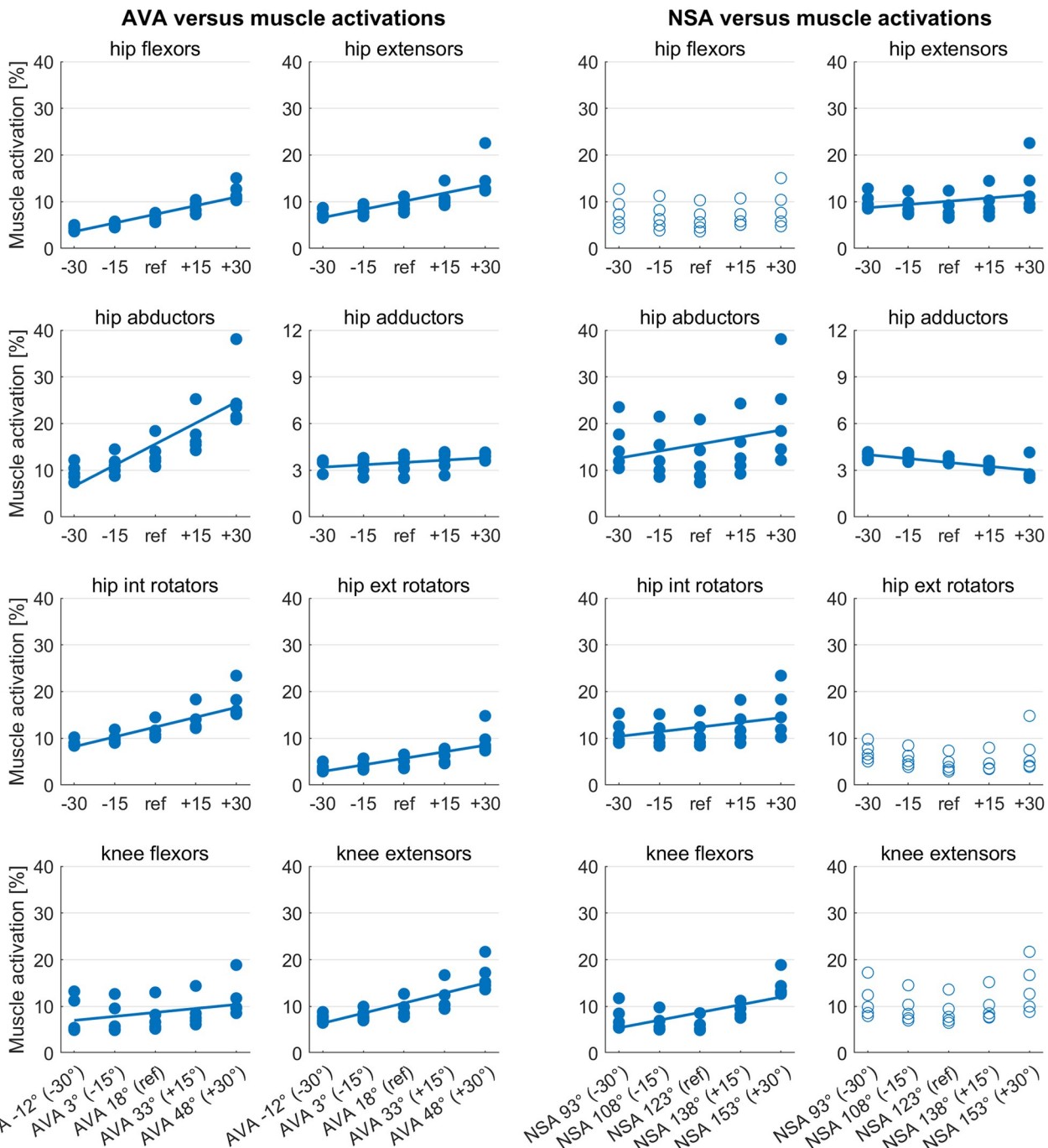

**Fig 6. Scatterplots showing the relationship between the femoral geometry (AVA and NSA) and mean muscle activations during the stance phase of the gait cycle.** Filled circles indicate significant predictors (p<0.05) from the multiple regression analysis. Straight lines are the regression lines obtained from the multiple regression analysis.

extensor and flexor muscle moments, which resulted in increased knee joint contact forces in models with increasing AVA and NSA.

Hip and knee joint contact forces significantly increased with increasing AVA and NSA in our study, which confirmed the findings of previous studies [20–22, 24]. Our study, however,

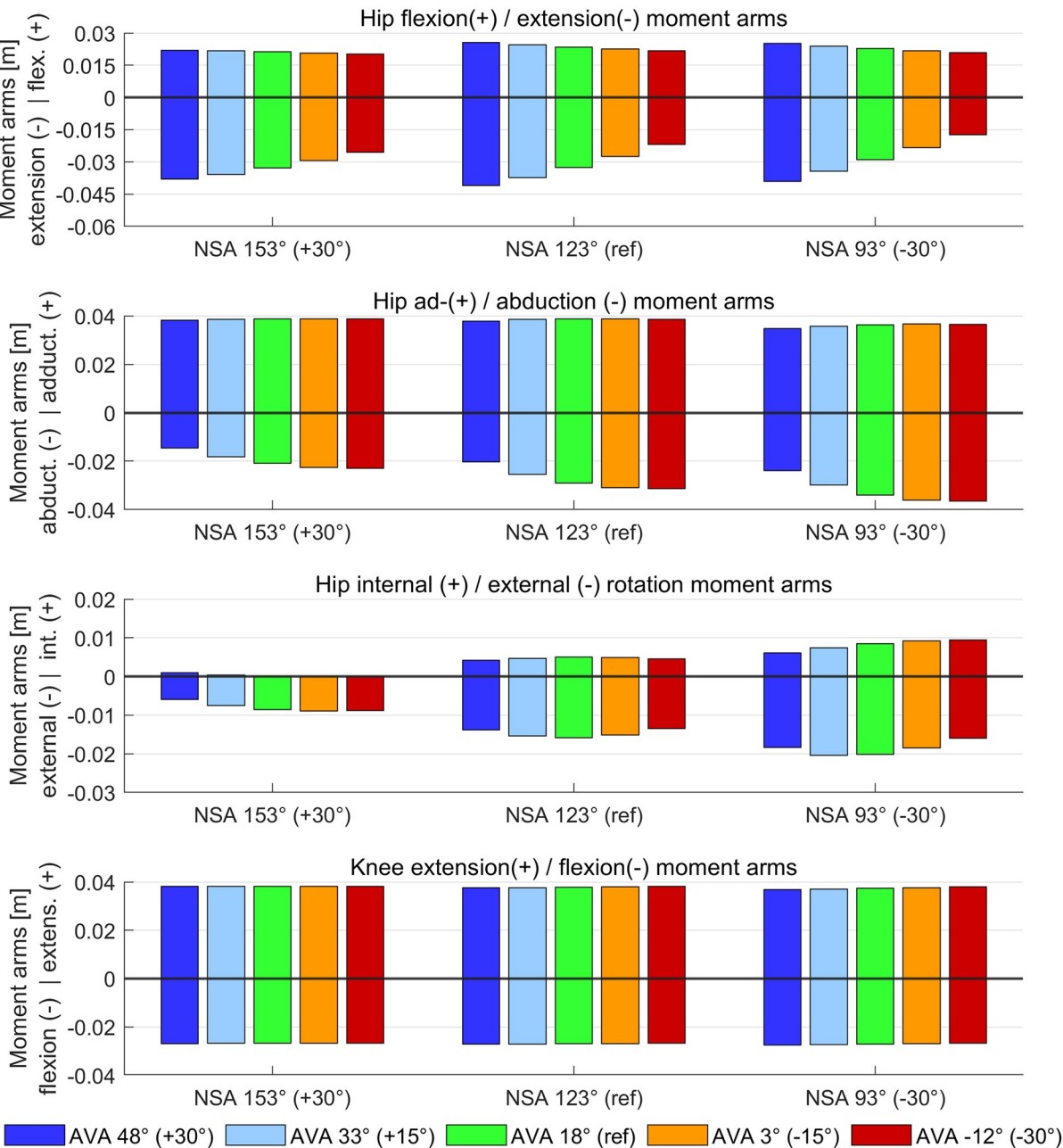

**Fig 7. Average moment arm during the stance phase of gait obtained from agonist and antagonist muscle groups.** Moment arms of agonist and antagonist are visualized in the same subplots as bar plots with either positive or negative values, respectively. Muscles were grouped based on their average moment arm in each anatomical plane during the stance phase of the gait cycle.

was the first study that investigated the combined impact of altered AVA and NSA on muscle forces and joint loads. The highest joint contact forces were observed in the model with 30˚-increased AVA and NSA, i.e. AVA of 48˚ and NSA of 153˚. Hip and knee joint contact forces in this model increased more than five times body weight compared to the values of the reference model (Fig 2). Interestingly, only in models with AVA above 18˚, increasing and decreasing NSA increased knee joint contact forces. These findings highlight that it is important to

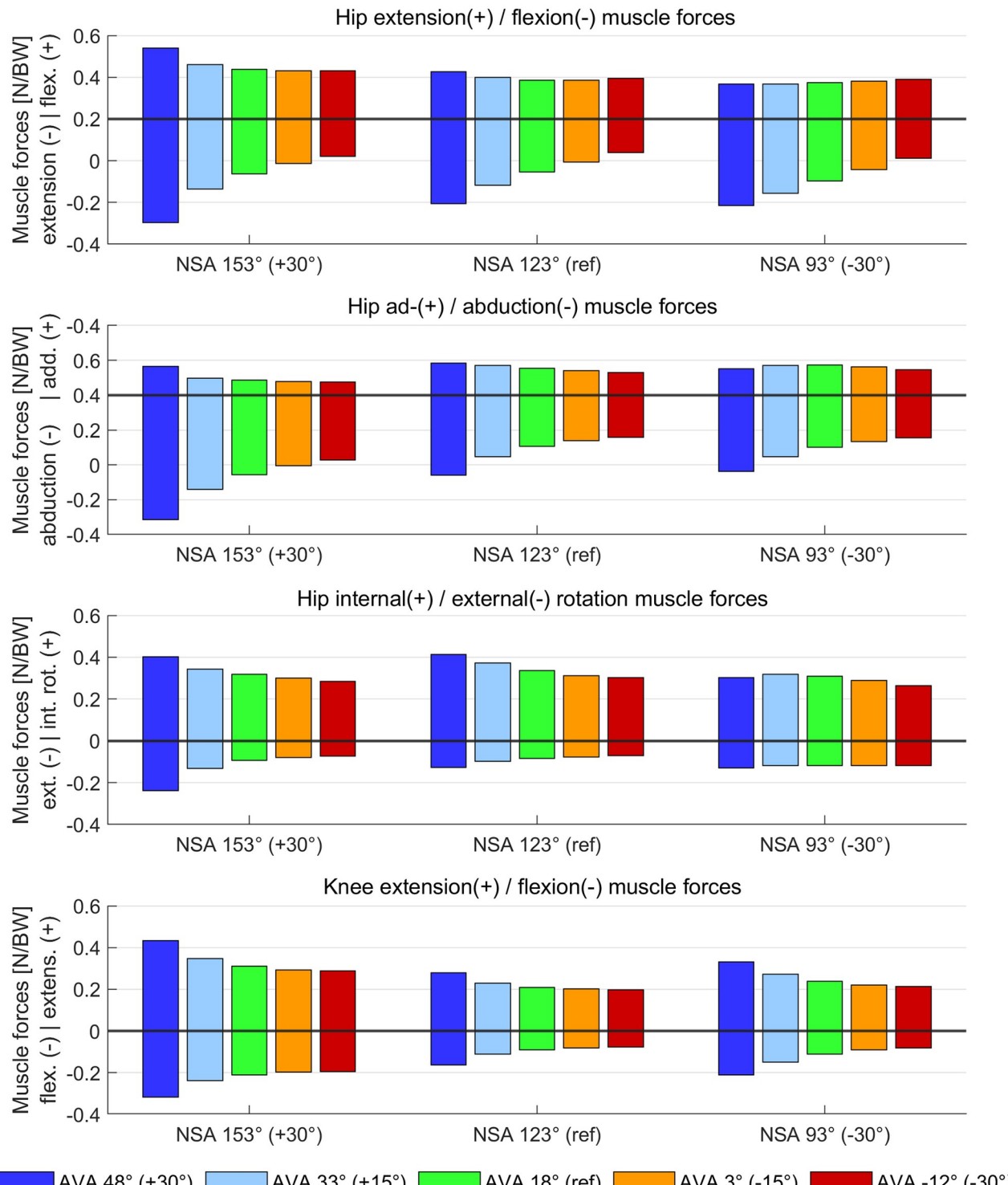

**Fig 8. Average muscle forces during the stance phase of gait obtained from agonist and antagonist muscle groups.** Muscle forces of agonist and antagonist are visualized in the same subplots as bar plots with either positive or negative values, respectively. Muscles were grouped based on their average moment arm in each anatomical plane during the stance phase of the gait cycle (same as in Fig 7).

account for both, the subject-specific AVA and NSA, when estimating joint contact forces. Neglecting the NSA or AVA, as previously done in some studies [22, 26], might lead to errors up to a magnitude of five times body weight and therefore could lead to misleading interpretations.

Increasing AVA resulted in a significant increase in co-contraction and muscle moments of hip and knee flexor and extensor muscles, whereas in other hip planes only minor alterations in muscle moments were observed. The increased co-contraction of hip and knee flexor/extensor muscles can be explained by the following cascade:

1. Increased AVA decreased the hip abduction moment arms of the majority of hip abductor muscles (Fig 7), i.e. gluteus minimus, medius and maximus, tensor fasciae, piriformis, which was in agreement with a previous simulation study [24]. Hence, more muscle forces had to be generated for the majority of hip abductor muscles (Fig 8) to produce the required hip abduction moment. The increase in hip abductor muscle activations and forces did not increase muscle moments due to the decreased moment arms.

2. The mean hip extension moment arms of the gluteus minimus, medius, and maximus increased with increasing AVA. Considering that the gluteus muscles were increasingly activated to compensate for the reduced hip abduction moment arms, the increased gluteus muscle forces combined with the increased hip extension moment arms produced an increased hip extension moment, which had to be counterbalanced by the antagonist muscles, i.e. hip flexor muscles. Hip flexor moment arms were barely altered and therefore several hip flexor muscles, i.e. rectus femoris, psoas and iliacus, were increasingly activated and produced higher muscle forces and therefore higher muscle moments and co-contraction with increasing AVA.

3. The rectus femoris, psoas, and iliacus are the strongest hip flexor muscles, i.e. have the largest isometric muscle forces compared to all other hip flexor muscles, and therefore their forces were increased to generate the required hip flexion muscle moment. The rectus femoris is, however, also a knee extensor muscle and therefore produced an additional knee extension moment. This extension moment was counterbalanced by an additional knee flexion moment produced mainly by increased muscle activations and forces of the gastrocnemius (medial and lateral) and short head of the biceps femoris muscles.

4. The increased co-contraction at the hip and knee-spanning muscle highlighted above explains the increased hip and knee joint contact forces in models with increased AVA. Additional figures of moment arms and muscle forces of all individual muscles are provided in the supplementary material and confirm the presented cascade leading to the increased co-contractions.

Large NSA combined with AVA of 48˚ increased both peaks of the hip joint contact force. Considering that hip flexor/extensor muscle moments did not change, hip ab-/adductor muscle moments only increased during the second half of the stance phase and hip internal/external rotator muscle moments decreased with increasing NSA, muscle moments of hip-spanning muscles cannot fully explain the observed increase in hip joint contact forces. Increasing NSA decreased the moment arm of many agonist and antagonist hip muscles and several muscles (e.g. gluteus medius and maximus) had to produce higher muscle forces to maintain the prescribed joint moments, which likely explains the observed increases in both peaks of hip joint contact forces. In models with large AVA, NSA angles of 153˚ increased the hip flexor and extensor muscle moments especially during the first half of the stance phase. This additional hip flexor/extensor co-contraction explains the big impact on the first peak of

hip joint contact forces in models with increased AVA and NSA (see figures in supplementary material).

Increasing NSA increased knee joint contact forces. Increases in the second peak of knee joint contact forces with increasing NSA were observed in models with low AVA (18˚ or less) while, in models with large AVA (48˚), both peaks of the knee joint contact force increased with increasing NSA from 138˚ to 153˚. The timely concurrent observed increase in knee flexor and extensor muscle moment, i.e. only during the second half of the stance phase in models with low AVA and during the first and second half of the stance phase in models with large AVA, confirm the increased amount of co-contraction responsible for the increased knee joint contact forces (see figures in supplementary material). The rectus femoris and gastrocnemius muscles produced higher forces and were mainly responsible for the increased knee flexor and extensor muscle moments. It seems that the rectus femoris muscle, whose moment arms are not altered with increasing NSA and AVA, has to generate greater forces to compensate for the reduced muscle moments of several hip-spanning muscles due to reduced moment arms.

External joint moments are often used as a surrogate measure for joint contact forces [51, 52]. Joint moments estimated via inverse dynamics consider the kinematics of the person and external forces, i.e. ground reaction forces, whereas joint contact forces additionally depend on muscle forces, which are influenced by each muscle's moment arm and line of action. Our study highlighted that large differences in joint contact forces, i.e. more than 100% of the reference values or five times body weight, can be caused by different femoral geometries even if joint kinematics and joint moments are exactly the same. Taking into account the large variability in femoral geometry in children as well as in adults (95% confidence intervals of approximately ±10˚ and ±20˚ for NSA and AVA, respectively [53–55]), we suggest not to use joint moments as a surrogate measure for joint contact forces. Furthermore, findings from studies that calculated correlations between joint moments and joint contact forces estimated with musculoskeletal models with generic bones [44, 56] should be interpreted with caution.

Many people with altered femoral geometry walk with a pathological gait pattern [2–4, 57–59], which was not considered in our study. Nevertheless, our simulation results might help to explain why some people do not use a typical walking pattern. Increasing AVA in our models led to substantial higher muscle activations for the majority of hip and knee muscle groups (Fig 6), which is unlikely to be sustained over a long period. In a pilot study with a person with increased AVA we showed that the patient-specific in-toeing gait pattern can reduce muscle activations and joint contact forces to typical values [60]. This is in agreement with Alexander et al. [26] who recently showed that patients with idiopathic increased AVA and in-toing gait do not have increased hip and knee loads. In children with cerebral palsy and increased AVA, Bosmans et al. [19] showed that patient-specific gait patterns reduce hip joint loading. Hence, it seems that, in many people with increased AVA, a pathological gait pattern is used to decrease muscular effort and potentially avoid pain due to increased joint contact forces.

We only assessed the influence of femoral geometry on muscle and joint contact forces and kept the tibia the same in all models. Torsional deformities, however, often occur concurrent at the femur and tibia, which might impact gait patterns and joint loading. Future studies should, therefore, investigate how combined torsional deformities at the femur and tibia influence muscle and joint contact forces.

Higher muscle forces, especially of the hip abductor and flexor muscles, were required to produce the necessary joint moments in models with increased AVA compared to the reference model (Fig 8). This indicates that it might be challenging or even impossible for people with hip muscle weakness and increased AVA to walk with a typical gait pattern. A recent study [39] showed that patients with increased AVA and in-toeing gait require less gluteus

medius muscle forces compared to a healthy control group. In patients with increased AVA but normal foot progression angle, Passmore et al. [25] showed that higher gluteus medius muscle forces are needed for walking with the patient-specific femoral geometry compared to a normal, typical geometry. Several other studies [61–64] investigated the relationship between femoral geometry and muscle strength. To the best of our knowledge, no studies showed a clear relationship between the femoral geometry (combined AVA and NSA), muscle strength and gait pattern. Hence, further research is needed to determine how the patient-specific femoral geometry influences the patient's gait pattern and vice versa.

Low joint contact forces were observed in models with decreased AVA and NSA. It is important to highlight that muscle and joint contact forces only account for a small portion of all the clinically-relevant parameters. Many other factors, e.g. labral tear [65] and femoroacetabular impingement [66], could add to clinical problems in people with decreased NSA and/ or reduced AVA.

We calculated muscle forces using static optimization, which might underestimate muscle co-contraction and joint contact forces. Hence, our simulations showed the minimum required co-contraction for a certain AVA and NSA combination. In other words, our findings showed that you cannot avoid muscle co-contraction if you have large AVA and NSA, and walk with a normal gait pattern. We used static optimization because it is the most common used method to estimate muscle forces and it has been shown to work very well for gait simulations compared to other optimization techniques [67]. Nevertheless, several studies showed that calibrating muscle parameters based on electromyography (EMG) signals and using an EMG-informed musculoskeletal model increases the accuracy of the simulations [40, 68–72], especially in people with neurological disorders [50, 73]. In our study based on 'what-if' simulations it was not possible to calibrate muscle parameters based on EMG data and therefore the absolute values of our estimated muscle and joint contact forces should be interpreted with caution.

Considering that abnormal joint loads are a primary risk factor for the development and progression of osteoarthritis [74], our findings might help to explain previous clinical observations. A recent systematic review [8] showed that increased AVA is associated with earlier and more severe hip osteoarthritis, whereas decreased AVA, i.e. retroversion, did show inconsistent results and no strong correlation with the development of osteoarthritis. In contrast, early work from Tönnis and Heinecke [75] suggested that diminished AVA can cause pain and osteoarthritis. In their paper, 17% of patients with diminished AVA had high NSA (above 140˚). A study including 111 patients with idiopathic knee osteoarthritis showed that NSA beyond 134˚ (up to 148˚) increases the risk of severe osteoarthritis eightfold [76]. We found increased hip and knee joint loads in models with increased AVA and NSA but not with decreased AVA, which might explain the observed association between hip osteoarthritis and increased AVA [8] and knee osteoarthritis and increased NSA [76]. Based on the findings of the systematic review [8] and our simulation results, we assume that the increased NSA and not the reduced AVA caused pain and osteoarthritis in some patients from Tönnis and Heinecke [75].

A recent study [4] reported high incidence of hip (63%) and knee (58%) joint pain in children with idiopathic increased AVA. In that study [4], many children with idiopathic increased AVA walked with an in-toeing gait but some children walked with foot progression angles comparable to the control group, which made a comparison with our findings possible. We found increased hip and knee joint contact forces in models with large AVA, which suggest a link between large contact forces and the incidence of pain in these two joints. Importantly, in-toeing gait might completely change the loading situation as shown in previous studies [26, 60].

Derotation osteotomies are used to correct femoral and tibial malrotations with the aim to improve the patients' gait pattern and avoid degenerative joint diseases in the long run [77, 78]. Our findings suggest that the clinical team should include the quantification and consideration of the NSA when planning derotation osteotomies. In people with large NSA, solely the derotation of the femur might not lead to a reduction in joint loads in the presence of a typical gait pattern.

A recent simulation study [23] investigated the impact of femoral AVA on patellofemoral joint loads. The authors found increased patellofemoral loads when AVA was increased from 22° to 42°, whereas a decrease in joint load was observed for the model with an AVA of 52°. This study [23] implemented a different modelling approach, i.e. muscle-driven simulations, which also altered joint kinematics and kinetics and therefore makes a direct comparison with our findings difficult. Nevertheless, the rectus femoris was found to greatly influence patellofemoral loads with altered AVA [23]. This is in agreement with the observed increases in rectus femoris forces in our models with large AVA, which led to increased knee joint contact forces.

Muscle activations of the reference model led to unrealistic high values. Hence, we had to multiply the maximum isometric muscle forces of the reference model by a factor of two to generate realistic muscle activation waveforms. The unrealistic high muscle activations were probably caused by a combination of the large body weight (73.1 kg) and fast walking velocity (1.41 m/s) of our participant [79]. Furthermore, we used the gait2392 model, which is known to include relatively low isometric muscle forces, which might not be representative for younger people. Newer models, e.g. Rajagopal model [80], include more realistic muscle properties (e.g. 6194 N instead of 3549 N for maximum isometric forces of the soleus). We, however, could not use the Rajagopal model for our study because the Torsion Tool [29] enables the modification of the AVA and NSA only for the gait2392 model. We plan to update the Torsion Tool in the near future to enable the personalization of the femoral and tibial geometries in all OpenSim models.

Our study included the following limitations. First, we only included motion capture data of one healthy participant, whereas the gait pattern, especially in people with symptomatic femoral deformities, is very heterogeneous [3, 58, 59, 78]. Our musculoskeletal modelling study, however, enabled us to comprehensively quantify how muscle and joint contact forces are altered solely due to changes in femoral geometry without any confounding factors. Future studies can extend our investigation and evaluate how the patient-specific gait patterns influence muscle and joint contact forces. Second, our musculoskeletal model included a planar knee model [81], which did not allow any out of sagittal plane movements. Furthermore, we locked the subtalar joint in our models due to an insufficient number of markers to track both the talocrural and subtalar joints. Considering that our simulations were based on gait data of a healthy person and subtalar joint movements are minor compared to talocrural movements [82], we believe that these limitations had a neglectable impact on our simulation results. Third, cartilage stress is associated with joint pain and osteoarthritis [83, 84], whereas we only estimated joint contact forces. Detailed cartilage stress analyses based on finite element models were beyond the scope of our study. Future studies, however, can use our musculoskeletal simulation results (freely available on https://simtk.org/projects/bone_gait_load) as input for further biomechanical investigations to quantify cartilage stress. Fourth, we did not assess how femoral anteversion deformities at different locations of the femur influence muscle and joint contact forces.

In conclusion, we conducted a musculoskeletal modelling study and showed how altered proximal femoral geometries influence muscle and joint contact forces during a typical gait pattern. We showed that increased joint contact forces in models with increased AVA and NSA are linked to changes in hip muscle moment arms and compensatory increases in hip

and knee muscle forces. Our findings might help to explain clinical observations, e.g. pain, degenerative joint diseases, and why a typical gait pattern is problematic in some patients with femoral deformities.

## Supporting information

**S1 File. Additional figures and table to support the findings and conclusion of our study.** (DOCX)

**S2 File. Verification of model modifications and simulation results.** (DOCX)

## Acknowledgments

We would like to thank Dr. Basilio Goncalves for his constructive feedback on the first draft of our manuscript.

## Author Contributions

**Conceptualization:** Hans Kainz, Gabriel T. Mindler, Andreas Kranzl.

**Data curation:** Andreas Kranzl.

**Formal analysis:** Hans Kainz.

**Investigation:** Hans Kainz, Gabriel T. Mindler, Andreas Kranzl.

**Methodology:** Hans Kainz.

**Project administration:** Hans Kainz.

**Resources:** Hans Kainz.

**Validation:** Hans Kainz.

**Visualization:** Hans Kainz.

**Writing – original draft:** Hans Kainz, Gabriel T. Mindler.

**Writing – review & editing:** Hans Kainz, Gabriel T. Mindler, Andreas Kranzl.

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
