## [Decision Letter · Decision Letter 0]

26 Jun 2023

PONE-D-23-11896Influence of femoral anteversion angle and neck-shaft angle on muscle forces and joint loading during walkingPLOS ONE

Dear Dr. Kainz,

Thank you for submitting your manuscript to PLOS ONE. After careful consideration, we feel that it has merit but does not fully meet PLOS ONE’s publication criteria as it currently stands. Therefore, we invite you to submit a revised version of the manuscript that addresses the points raised during the review process.

The Reviewers have raised a number of points summarised in the review template and also in teh attached version of your manuscript with comments. 

We look forward to receiving your revised manuscript.

Kind regards,

Rory O'Sullivan, PhD

Academic Editor

PLOS ONE

Reviewers' comments:

Reviewer's Responses to Questions

**Comments to the Author**

1. Is the manuscript technically sound, and do the data support the conclusions?

Reviewer #1: Partly

Reviewer #2: Yes

2. Has the statistical analysis been performed appropriately and rigorously? 

Reviewer #1: I Don't Know

Reviewer #2: Yes

3. Have the authors made all data underlying the findings in their manuscript fully available?

Reviewer #1: No

Reviewer #2: Yes

4. Is the manuscript presented in an intelligible fashion and written in standard English?

Reviewer #1: Yes

Reviewer #2: Yes

5. Review Comments to the Author

Reviewer #1: Manuscript number: PONE-D-23-11896

General Comments to the Authors

This manuscript describes a modeling perturbation study in which the authors varied the femoral anteversion angle (AVA) and neck-shaft angle (NSA) in large increments and calculated how joint contact forces and muscle forces changed. The authors found that increasing NSA and AVA both had significant effects on hip and knee joint contact forces. The study is a classic “what-if” type of perturbation that adds some new information to the literature about the influence of NSA on muscle and joint contact forces. Information about changes to the AVA does not provide anything new or insightful beyond the many studies that have already been published. The authors did cite many of the previous studies, but a few are missing that came up with a search of literature databases. While the AVA information is not new, understanding AVA changes in combination with NSA changes is useful.

The study has some limitations that, although being mentioned by the authors, nevertheless reduce the impact of the study. First, the authors chose to use static optimization to solve for muscle forces. The authors recognize that static optimization poorly estimates co-contractions, yet proceeded to make co-contractions a major theme of the study. I actually think that the co-contraction analysis used by the authors is good and could be informative, but it depends on reasonable estimation of co-contracting muscle forces, which static optimization does not do well. The authors state that because static optimization is unable to estimate co-contractions, readers should assume that there is much more co-contraction happening than is reported. What extrapolations on the results should readers make with this assumption? More co-contraction could increase or decrease joint contact loads, so the readers cannot really know what the current study gets right or wrong and what can guessed beyond the study results. Why not use a different solver that can better estimate co-contraction?

Second, why the use of gait2392? The authors recognize that muscle parameters in gait2392 do not reflect the parameters of their sole participant and they needed to adjust maximum isometric forces by a factor of two. While it is common for modelers to increase the maximum isometric force when modeling tasks like running, many studies have used the baseline model successfully for walking simulations. Why did the baseline model fail here? That seems odd. Was there something unusual about the gait data that were used? Also, why not use a model that has been adapted for younger people, like that of Rajagopal et al?

Third, the Introduction and Discussion are both very long and repeat themselves in several places. Such lengthy sections make the paper more difficult to read with interest. Can you find ways to reduce duplicated statements and ideas?

Specific Comments

Line 51 – If you note that the AVA is a measure in the transverse plane, it is worth noting that the NSA is a coronal plane measure.

Line 78 and Line 86 – Referring to prior work as Scot Delp’s or Ilsa Jonkers’ group diminishes the contributions of the first authors. Please cite the specific papers or do not call out specific labs.

References: The paper is well referenced. Two references that are very much related are

- Shepherd MC, Gaffney BMM, Song K, Clohisy JC, Nepple JJ, Harris MD. Femoral version deformities alter joint reaction forces in dysplastic hips during gait. J Biomech. 2022 Apr;135:111023.

- Lerch TD, Eichelberger P, Baur H, Schmaranzer F, Liechti EF, Schwab JM, Siebenrock KA, Tannast M, 2019. Prevalence and diagnostic accuracy of in-toeing and out-toeing of the foot for patients with abnormal femoral torsion and femoroacetabular impingement: Implications for hip arthroscopy and femoral derotation osteotomy. Bone Joint J 101-B (10), 1218–1229. 10.1302/0301-620X.101B10.BJJ-2019-0248.R1

While those two references appear to be a slightly different patient group than those targeted in the current paper, Shepherd reported hip contact force changes with AVA changes while holding subject-specific gait patterns constant (like the current study), and Lerch paper reported that patients with abnormal femoral version can have outwardly normal gait patterns (similar to Passmore et al).

Introduction – The Intro provides a good summary of much that has been done to investigate AVA and muscle forces or contact forces, but it doesn’t lead naturally to the study objective. For example, the emphasis given to variability in gait patterns within the Intro, makes one think that the current study would address that variability – which it does not. Can you revise to shorten the Intro and lead more directly to the study objective?

Methods: musculoskeletal models: As written it is not clear that AVA and NSA are varied together. By adding up the number of models and looking at the results, the reader can figure it out, but it might help to explicitly state that the two variables were varied separately and together (or just together?).

Line 145: Why is the date of gait data collection or modeling relevant for this study of a single subject? I think you can remove this information unless the journal explicitly asks.

Line 145: Was the subject chosen at random? If not, why was this subject selected for analysis?

Line 172: I think the word “as” is missing between to and hip.

Lines 172-173: It makes sense to group the muscle moments as ‘hip flexors and extensors’, etc, but how did you group the actual muscles when analyzing co-contractions? Were muscles assigned to a single group or could they exist in multiple groups and if so, did that grouping vary throughout the gait cycle? Did you simply use the grouping listed in the gait 2392 model?

Lines 189-191 and 326-334: It is unnecessary to include “results” that the joint angles and moments did not change. Based on how the model perturbations were performed, it is not expected that they would change. Thus, results and discussion are not needed. I suggest taking some of the language from the Discussion paragraph and moving it to the Methods to explain why joint angles and moments would not change with AVA and NSA changes.

Line 245 – I could not find data in the supplementary material that actually supports ‘reasonable agreement’ of model activations with experimental EMG signals. Also, is this agreement only true for the reference model?

Line 25-251: Where are the data supporting the verification of smooth waveforms throughout the gait cycle?

Line 347: I suggest caution with the overstatement about comprehensively describing the link between AVA and increased joint loads. Several previous studies have established this link. Also, it is dangerous to claim that a study is fully comprehensive. For instance, the current study does not explore the effect of femoral torsion changes that can occur at different regions of the femur. The current study adds the effect of NSA changes, which is useful.

Lines 367-376: I suggest that the statements about relative retroversion be softened and stick to the data available. A strong statement about retroversion not causing out-toeing cannot be made from the one subject and one gait pattern used in the current study.

Lines 377-385: This paragraph can be removed to shorten the Discussion. It was unclear what important takeaways this paragraph added.

Lines 402-403 and 408-409 are essentially the same sentence. Consider revising the paragraph to shorten.

Lines 415-439: The arguments about associations among AVA, NSA, and osteoarthritis seem to be a bit circular in this page. Consider revising and shortening to make a clear statement based on current and prior results.

Generally, the main takeaway messages of the study get diluted and lost with the excessively long Discussion, which seems to then necessitate two long conclusion paragraphs that restate what has already been said.

Reviewer #2: I have read this manuscript with great interest. It documents a very thorough and rigorously conducted study. The findings will be useful for the readers, and the limitations are clearly articulated. I have only added relatively minor comments, with the intention to improve the paper a bit more.

6. PLOS authors have the option to publish the peer review history of their article (what does this mean?). If published, this will include your full peer review and any attached files.

Reviewer #1: No

Reviewer #2: No

---

## [Author Response · Author response to Decision Letter 0]

13 Jul 2023

RESPONSE TO REVIEWERS’ COMMENTS

Journal: PLOS ONE

Manuscript Number: PONE-D-23-11896

Title: Influence of femoral anteversion angle and neck-shaft angle on muscle forces and joint loading during walking

We thank the Editor and Reviewers for reviewing our manuscript and are thankful for the feedback provided by the Reviewers. Please find below the Reviewers’ comments and Authors’ responses. All changes to the manuscript are shown in red font in the text below.

Reviewer #1

Comment reviewer #1: 

General Comments to the Authors

This manuscript describes a modeling perturbation study in which the authors varied the femoral anteversion angle (AVA) and neck-shaft angle (NSA) in large increments and calculated how joint contact forces and muscle forces changed. The authors found that increasing NSA and AVA both had significant effects on hip and knee joint contact forces. The study is a classic “what-if” type of perturbation that adds some new information to the literature about the influence of NSA on muscle and joint contact forces. Information about changes to the AVA does not provide anything new or insightful beyond the many studies that have already been published. The authors did cite many of the previous studies, but a few are missing that came up with a search of literature databases. While the AVA information is not new, understanding AVA changes in combination with NSA changes is useful.

Authors’ response:

We are grateful for the constructive feedback provided by the reviewer. We agree with the reviewer that some papers already showed the increase in joint loads with increasing AVA. In agreement with the reviewer, we believe that our comprehensive analyses provide new insights and useful information about the NSA and combination of AVA and NSA on muscle and joint contact forces. Furthermore, the mechanism how increased AVA influence knee loads has not been described previously. Hence, we believe our paper will be of high interest for the research as well as clinical community.

Comment reviewer #1: 

The study has some limitations that, although being mentioned by the authors, nevertheless reduce the impact of the study. First, the authors chose to use static optimization to solve for muscle forces. The authors recognize that static optimization poorly estimates co-contractions, yet proceeded to make co-contractions a major theme of the study. I actually think that the co-contraction analysis used by the authors is good and could be informative, but it depends on reasonable estimation of co-contracting muscle forces, which static optimization does not do well. The authors state that because static optimization is unable to estimate co-contractions, readers should assume that there is much more co-contraction happening than is reported. What extrapolations on the results should readers make with this assumption? More co-contraction could increase or decrease joint contact loads, so the readers cannot really know what the current study gets right or wrong and what can guessed beyond the study results. Why not use a different solver that can better estimate co-contraction?

Authors’ response:

We partly agree with the reviewer. Different solvers might lead to slightly different results. However, we do not know which solver would lead to the most accurate results due to our ‘what-if’ simulation study design. We chose static optimization because it is the most used solver in the research community and it has been shown that static optimization works very well for gait simulations (Wesseling et al. 2014). We believe this is not a big limitation of our study due to the following reasons:

(1) We were mainly interested in the relative comparison of muscle and joint contact forces between different models and not the absolute values

(2) We did not investigate pathological gait pattern or patients with neurological disorders

(3) In general, people prefer to walk with a minimum amount of co-contraction

(4) It is very unlikely that more co-contraction will decrease joint contact forces 

 Based on the reviewer’s comment we added the following information to the discussion section of our manuscript:

‘We calculated muscle forces using static optimization, which might underestimate muscle co-contraction and joint contact forces. …. We used static optimization because it is the most common used method to estimate muscle forces and it has been shown to work very well for gait simulations compared to other optimization techniques [73]. Nevertheless, several studies…’

Wesseling M, Derikx LC, De Groote F, Bartels W, Meyer C, Verdonschot N, et al. Muscle optimization techniques impact the magnitude of calculated hip joint contact forces. J Orthop Res. 2015;33: 430–438. doi:10.1002/JOR.22769

Comment reviewer #1: 

Second, why the use of gait2392? The authors recognize that muscle parameters in gait2392 do not reflect the parameters of their sole participant and they needed to adjust maximum isometric forces by a factor of two. While it is common for modelers to increase the maximum isometric force when modeling tasks like running, many studies have used the baseline model successfully for walking simulations. Why did the baseline model fail here? That seems odd. Was there something unusual about the gait data that were used? Also, why not use a model that has been adapted for younger people, like that of Rajagopal et al?

Authors’ response:

We totally agree with the reviewer. The Rajagopal model would be a better model and should become the standard model for gait simulations in our community. The Torsion Tool (Veerkamp et al., 2021), which we used for modifying the femoral geometry, only worked with the gait2392 model. One other tool by Modenese et al. (2021) is available to modify the femur but it only allows to change the AVA and not the NSA and therefore it was not useful for the purpose of our study. Hence, we decided to use the gait2392 model for our simulations. Since submitting the paper, my research group updated the torsion tool, which should now work for all OpenSim models. We recently submitted a paper in which we introduce the updated tool and we already uploaded the updated scripts on simtk.org. However, we only developed and started using the updated tool after we conducted this study. Using a different model will very unlikely change the conclusion and main message of our study. Hence, we would prefer not to re-run all analysis and update all figures. However, if the reviewer persists, we would be willing to re-run all our analyses. Considering that two of my students worked on the updated torsion tool, we would also need to include two additional co-authors.

The second point raised by the reviewer is very interesting. We believe there are two factors which led to this apparently uncommon observation. (1) Our participant was quite heavy (73 kg) and walked very fast (1.4 m/s) compared to the average population (Schimpl et al., 2011). The combination of the high body weight and fast walking velocity is likely the reason for the high muscle activity of some muscles in the gait2392 model with the generic isometric muscle forces. (2) We assume that not all people check the muscle activations of their simulation. In OpenSim, the simulations do not fail and do not provide any error messages if muscles are activated 100%. It would, therefore, still be possible to calculate joint contact forces. 

If the reviewer wants, we could add the following paragraph to our discussion section:

‘Muscle activations of the reference model led to unrealistic high values. Hence, we had to multiply the maximum isometric muscle forces of the reference model by a factor of two to generate realistic muscle activation waveforms. The unrealistic high muscle activations were probably caused by a combination of the large body weight (73.1 kg) and fast walking velocity (1.41 m/s) of our participant [85]. Furthermore, we used the gait2392 model, which is known to include relatively low isometric muscle forces, which might not be representative for younger people. Newer models, e.g. Rajagopal model [86], include more realistic muscle properties (e.g. 6194 N instead of 3549 N for maximum isometric forces of the soleus). We, however, could not use the Rajagopal model for our study because the Torsion Tool [28] enables the modification of the AVA and NSA only for the gait2392 model. We plan to update the Torsion Tool in the near future to enable the personalization of the femoral and tibial geometries in all OpenSim models.’

We did not do it yet because one of the reviewer’s suggestion was to shorten the discussion section.

Veerkamp K, Kainz H, Killen BA, Jónasdóttir H, van der Krogt MM. Torsion Tool: An automated tool for personalising femoral and tibial geometries in OpenSim musculoskeletal models. J Biomech. 2021;125: 110589. doi:10.1016/J.JBIOMECH.2021.110589

Modenese L, Barzan M, Carty CP. Dependency of lower limb joint reaction forces on femoral version. Gait Posture. 2021;88: 318–321. doi:10.1016/J.GAITPOST.2021.06.014

Schimpl M, Moore C, Lederer C, Neuhaus A, Sambrook J, Danesh J, et al. Association between Walking Speed and Age in Healthy, Free-Living Individuals Using Mobile Accelerometry—A Cross-Sectional Study. PLoS One. 2011;6: e23299. doi:10.1371/JOURNAL.PONE.0023299

Comment reviewer #1: 

Third, the Introduction and Discussion are both very long and repeat themselves in several places. Such lengthy sections make the paper more difficult to read with interest. Can you find ways to reduce duplicated statements and ideas?

Authors’ response:

We are grateful for the reviewer’s feedback and agree that our manuscript was too long and repeated itself in several places. Based on the reviewer’s comment, we decreased the length of the manuscript by approximately 1,200 words. Both the length of the introduction and discussion sections were decreased by 25%.

Comment reviewer #1: 

Specific Comments

Line 51 – If you note that the AVA is a measure in the transverse plane, it is worth noting that the NSA is a coronal plane measure.

Authors’ response:

We agree with the reviewer that the NSA was traditionally only measured in the coronal plane. However, medical three-dimensional imaging enables a more accurate determination of the NSA (Bonneau et al. 2012; Sangeaux et al., 2015) and has become a standard in research. We therefore decided not to mention a specific plane for the measurement of the NSA. 

Bonneau, N. et al. A three-dimensional axis for the study of femoral neck orientation. J. Anat. 221, 465–476 (2012).

Sangeux, M., Pascoe, J., Kerr Graham, H., Ramanauskas, F. & Cain, T. Three-dimensional measurement of femoral neck anteversion and neck shaft angle. J. Comput. Assist. Tomogr. 39, 83–85 (2015).

Comment reviewer #1: 

Line 78 and Line 86 – Referring to prior work as Scot Delp’s or Ilsa Jonkers’ group diminishes the contributions of the first authors. Please cite the specific papers or do not call out specific labs.

Authors’ response:

We thank the reviewer for the feedback and agree with the comment. We modified this section and removed the names of the specific labs.

Comment reviewer #1: 

References: The paper is well referenced. Two references that are very much related are

- Shepherd MC, Gaffney BMM, Song K, Clohisy JC, Nepple JJ, Harris MD. Femoral version deformities alter joint reaction forces in dysplastic hips during gait. J Biomech. 2022 Apr;135:111023.

- Lerch TD, Eichelberger P, Baur H, Schmaranzer F, Liechti EF, Schwab JM, Siebenrock KA, Tannast M, 2019. Prevalence and diagnostic accuracy of in-toeing and out-toeing of the foot for patients with abnormal femoral torsion and femoroacetabular impingement: Implications for hip arthroscopy and femoral derotation osteotomy. Bone Joint J 101-B (10), 1218–1229. 10.1302/0301-620X.101B10.BJJ-2019-0248.R1

While those two references appear to be a slightly different patient group than those targeted in the current paper, Shepherd reported hip contact force changes with AVA changes while holding subject-specific gait patterns constant (like the current study), and Lerch paper reported that patients with abnormal femoral version can have outwardly normal gait patterns (similar to Passmore et al).

Authors’ response:

We thank the reviewer for the hint to these very interesting papers. We added the references several times throughout the manuscript, where it was appropriate.

Comment reviewer #1: 

Introduction – The Intro provides a good summary of much that has been done to investigate AVA and muscle forces or contact forces, but it doesn’t lead naturally to the study objective. For example, the emphasis given to variability in gait patterns within the Intro, makes one think that the current study would address that variability – which it does not. Can you revise to shorten the Intro and lead more directly to the study objective?

Authors’ response:

We agree with the reviewer. We shortened the introduction and removed the paragraph about gait variability.

Comment reviewer #1: 

Methods: musculoskeletal models: As written it is not clear that AVA and NSA are varied together. By adding up the number of models and looking at the results, the reader can figure it out, but it might help to explicitly state that the two variables were varied separately and together (or just together?).

Authors’ response:

Based on the reviewer’s comment we added the following information:

‘Both variables (AVA and NSA) were varied separately and together compared to the values of the reference model.’

Comment reviewer #1: 

Line 145: Why is the date of gait data collection or modeling relevant for this study of a single subject? I think you can remove this information unless the journal explicitly asks.

Authors’ response:

We agree with the reviewer and removed this information from the manuscript.

Comment reviewer #1: 

Line 145: Was the subject chosen at random? If not, why was this subject selected for analysis?

Authors’ response:

A random subject was chosen. We added the following information to this sentence.

‘AK selected a random data set for this study.’

Comment reviewer #1: 

Line 172: I think the word “as” is missing between to and hip.

Authors’ response:

We thank the reviewer for the hint and modified the sentence as suggested.

Comment reviewer #1: 

Lines 172-173: It makes sense to group the muscle moments as ‘hip flexors and extensors’, etc, but how did you group the actual muscles when analyzing co-contractions? Were muscles assigned to a single group or could they exist in multiple groups and if so, did that grouping vary throughout the gait cycle? Did you simply use the grouping listed in the gait 2392 model?

Authors’ response:

We defined the function of each muscle based on the moment arm of each muscle during each frame of the gait cycle. E.g. if a muscle had a hip flexor moment arm during the first 60% of the gait cycle, the muscle force of this muscle was multiplied with the moment arm during this period of the gait cycle. Hence, depending on the moment arm, one muscle could be part of different groups. We added the following additional information to our manuscript:

‘The functional role of each muscle was defined by the muscle’s moment arm during each frame of the gait cycle.’

Comment reviewer #1: 

Lines 189-191 and 326-334: It is unnecessary to include “results” that the joint angles and moments did not change. Based on how the model perturbations were performed, it is not expected that they would change. Thus, results and discussion are not needed. I suggest taking some of the language from the Discussion paragraph and moving it to the Methods to explain why joint angles and moments would not change with AVA and NSA changes.

Authors’ response:

We agree with the reviewer that this information is not needed for a person with a technical and/or musculoskeletal modelling background. We, however, believe that this is not clear for all readers from the journal and several people from the clinical gait analysis community. Based on the reviewer’s comment, we moved some of the information to the method section and deleted this paragraph from the discussion section.

Comment reviewer #1: 

Line 245 – I could not find data in the supplementary material that actually supports ‘reasonable agreement’ of model activations with experimental EMG signals. Also, is this agreement only true for the reference model?

Authors’ response:

Figure S19 and figure S20 in the supplementary material report the muscle activation and force waveforms from our simulations. Based on the reviewer’s comment, we modified the figure legends to refer to the papers which we used to qualitative compare our results with previously reported values. The modified models mainly changed the magnitude but not the timing of muscle activation and therefore barely changed the agreement with the previously reported data because we could only compare the timing and not the magnitude of the muscle activations with the EMG signals.

Comment reviewer #1: 

Line 25-251: Where are the data supporting the verification of smooth waveforms throughout the gait cycle?

Authors’ response:

We added the missing information to the supplementary material. See the last four pages in the supplementary material.

Comment reviewer #1: 

Line 347: I suggest caution with the overstatement about comprehensively describing the link between AVA and increased joint loads. Several previous studies have established this link. Also, it is dangerous to claim that a study is fully comprehensive. For instance, the current study does not explore the effect of femoral torsion changes that can occur at different regions of the femur. The current study adds the effect of NSA changes, which is useful.

Authors’ response:

We agree with the reviewer and deleted the whole paragraph. Furthermore, we added the reference from Shepherd et al. (2022) and mentioned the limitation of modelling femoral torsion only at one location of femur to the limitation section of our manuscript.

‘Fifth, we did not assess how femoral deformities at different locations of the femur influence muscle and joint contact forces.’

Comment reviewer #1: 

Lines 367-376: I suggest that the statements about relative retroversion be softened and stick to the data available. A strong statement about retroversion not causing out-toeing cannot be made from the one subject and one gait pattern used in the current study.

Authors’ response:

We agree with the reviewer. Our statement cannot be concluded from our study and therefore we deleted the first two sentences from this paragraph.

Comment reviewer #1: 

Lines 377-385: This paragraph can be removed to shorten the Discussion. It was unclear what important takeaways this paragraph added.

Authors’ response:

Based on the reviewer’s suggestion, we deleted this paragraph.

Comment reviewer #1: 

Lines 402-403 and 408-409 are essentially the same sentence. Consider revising the paragraph to shorten.

Authors’ response:

Based on the reviewer’s comment, we deleted one of these sentences and re-wrote the paragraph.

Comment reviewer #1: 

Lines 415-439: The arguments about associations among AVA, NSA, and osteoarthritis seem to be a bit circular in this page. Consider revising and shortening to make a clear statement based on current and prior results.

Authors’ response:

Based on the reviewer’s comment we re-wrote and shortened this paragraph.

Comment reviewer #1: 

Generally, the main takeaway messages of the study get diluted and lost with the excessively long Discussion, which seems to then necessitate two long conclusion paragraphs that restate what has already been said.

Authors’ response:

We agree with the reviewer and our discussion was too comprehensive. Based on the reviewer’s comment, we deleted one conclusion paragraph completely and shortened the second one. Furthermore, we decreased the length of the discussion section by 900 words.

Reviewer #2

Comment reviewer #2: 

I have read this manuscript with great interest. It documents a very thorough and rigorously conducted study. The findings will be useful for the readers, and the limitations are clearly articulated. I have only added relatively minor comments, with the intention to improve the paper a bit more.

Authors’ response:

We thank the reviewer for his/her constructive feedback and the nice words about our study. We implemented all the recommended changes in the revised manuscript, except of the following one.

Comment reviewer #2: 

This is a minor point, and my suggestion is optional.

The purpose of the in-toeing is internal hip rotation, which will improve the hip muscle moment arms. It would help the reader to refer to internal rotation gait, which is conceptually closer to the mechanism than the distal in-toeing.

Authors’ response:

In-toeing gait could be caused by torsional deformities and/or internal hip rotation. In other words, if in-toeing gait is mainly caused by the torsional deformities (hip internal/external rotation is the same as in typical/healthy people), the person might walk with in-toeing gait to avoid external hip rotation, which would be required to achieve a gait pattern with a typical foot progression angle. Internal hip rotation is only the reason for in-toeing gait if the femoral and tibial geometry is normal/typical in a person. Hence, we decided to keep the wording as it was in our manuscript.

---

## [Decision Letter · Decision Letter 1]

24 Aug 2023

PONE-D-23-11896R1Influence of femoral anteversion angle and neck-shaft angle on muscle forces and joint loading during walkingPLOS ONE

Dear Dr. Kainz,

Thank you for submitting your manuscript to PLOS ONE. After careful consideration, we feel that it has merit and you have addressed the review comments adequately.However, please note the re-review comment relating to the paragraph on muscle adaptations. If you decide not to include, please provide a brief justification.

We look forward to receiving your revised manuscript.

Kind regards,

Rory O'Sullivan, PhD

Academic Editor

PLOS ONE

Journal Requirements:

Reviewers' comments:

Reviewer's Responses to Questions

**Comments to the Author**

1. If the authors have adequately addressed your comments raised in a previous round of review and you feel that this manuscript is now acceptable for publication, you may indicate that here to bypass the “Comments to the Author” section, enter your conflict of interest statement in the “Confidential to Editor” section, and submit your "Accept" recommendation.

Reviewer #1: (No Response)

2. Is the manuscript technically sound, and do the data support the conclusions?

Reviewer #1: Yes

3. Has the statistical analysis been performed appropriately and rigorously? 

Reviewer #1: Yes

4. Have the authors made all data underlying the findings in their manuscript fully available?

Reviewer #1: Yes

5. Is the manuscript presented in an intelligible fashion and written in standard English?

Reviewer #1: Yes

6. Review Comments to the Author

Reviewer #1: Thank you for addressing all of my questions. Thank you for shortening the manuscript. If the authors are willing, I do believe it would be useful to add the paragraph they proposed in their response about muscle activations.

7. PLOS authors have the option to publish the peer review history of their article (what does this mean?). If published, this will include your full peer review and any attached files.

Reviewer #1: **Yes: **Michael Harris

---

## [Author Response · Author response to Decision Letter 1]

25 Aug 2023

Comment reviewer #1: 

Thank you for addressing all of my questions. Thank you for shortening the manuscript. If the authors are willing, I do believe it would be useful to add the paragraph they proposed in their response about muscle activations.

Authors’ response:

As suggested by the reviewer, we added the following paragraph to our discussion section:

‘Muscle activations of the reference model led to unrealistic high values. Hence, we had to multiply the maximum isometric muscle forces of the reference model by a factor of two to generate realistic muscle activation waveforms. The unrealistic high muscle activations were probably caused by a combination of the large body weight (73.1 kg) and fast walking velocity (1.41 m/s) of our participant [85]. Furthermore, we used the gait2392 model, which is known to include relatively low isometric muscle forces, which might not be representative for younger people. Newer models, e.g. Rajagopal model [86], include more realistic muscle properties (e.g. 6194 N instead of 3549 N for maximum isometric forces of the soleus). We, however, could not use the Rajagopal model for our study because the Torsion Tool [28] enables the modification of the AVA and NSA only for the gait2392 model. We plan to update the Torsion Tool in the near future to enable the personalization of the femoral and tibial geometries in all OpenSim models.’

---

## [Editor Report · Decision Letter 2]

30 Aug 2023

Influence of femoral anteversion angle and neck-shaft angle on muscle forces and joint loading during walking

PONE-D-23-11896R2

Dear Dr. Kainz,

We’re pleased to inform you that your manuscript has been judged scientifically suitable for publication and will be formally accepted for publication once it meets all outstanding technical requirements.

Kind regards,

Rory O'Sullivan, PhD

Academic Editor

PLOS ONE
---

## [Editor Report · Acceptance letter]

3 Oct 2023

PONE-D-23-11896R2 

Influence of femoral anteversion angle and neck-shaft angle on muscle forces and joint loading during walking 

Dear Dr. Kainz:

I'm pleased to inform you that your manuscript has been deemed suitable for publication in PLOS ONE. Congratulations! Your manuscript is now with our production department. 

Kind regards, 

on behalf of

Dr. Rory O'Sullivan 

Academic Editor

PLOS ONE